# Thyroid Function and Morphology in Gaucher Disease: Exploring the Endocrine Implications

**DOI:** 10.3390/ijms252413636

**Published:** 2024-12-20

**Authors:** Małgorzata Kałużna, Ewelina Szczepanek-Parulska, Jerzy Moczko, Magdalena Czlapka-Matyasik, Katarzyna Katulska, Katarzyna Ziemnicka, Beata Kieć-Wilk, Marek Ruchała

**Affiliations:** 1Ward of Endocrinology, Metabolism and Internal Diseases Ward, University Clinical Hospital, 60-786 Poznan, Poland; ewelinaparulska@gmail.com (E.S.-P.); kaziem@ump.edu.pl (K.Z.); mruchala@ump.edu.pl (M.R.); 2Department of Endocrinology, Metabolism and Internal Medicine, Poznan University of Medical Sciences, 60-355 Poznań, Poland; 3Department of Computer Science and Statistics, Poznan University of Medical Sciences, 60-355 Poznan, Poland; jmoczko@ump.edu.pl; 4Department of Human Nutrition and Dietetics, Faculty of Food Science and Nutrition, Poznan University of Life Sciences, 60-624 Poznan, Poland; magdalena.matyasik@up.poznan.pl; 5Department of Radiology, Poznan University of Medical Sciences, 60-355 Poznan, Poland; kkatulska@ump.edu.pl; 6Department of Pathophysiology, Jagiellonian University Medical College, St. John Paul II Socialist Hospital in Krakow, 31-121 Krakow, Poland; mbkiec@gmail.com

**Keywords:** Gaucher disease (GD), thyroid, hypothyroidism, nodular goiter, chitotriosidase, glucosylsphingosine (lysoGb-1), metabolism, glucose, elastography, shear-wave elastography (SWE)

## Abstract

Gaucher disease (GD), the most common ultra-rare metabolic disorder, results from lipid accumulation. Systemic inflammation, cellular stress, and metabolic dysfunction may influence endocrine function, including the thyroid. This study evaluated thyroid function and morphology in 60 GD patients, alongside carbohydrate and lipid metabolism. Anthropometric, biochemical, and hormonal tests were conducted, including thyroid ultrasound and shear-wave elastography (SWE). Clinical data, bone mineral density (BMD), and body composition (BOD POD) analyses were correlated. Healthy controls, matched for age, sex, and body mass index (BMI), were included. GD patients had higher thyroid stimulating hormone (TSH) and free thyroxine (FT4) levels within normal limits. Hypothyroidism occurred in 7%, elevated anti-thyroid antibodies in 8%, and nodular goiter in 23%. Patients with nodular goiter showed lower platelet counts and higher chitotriosidase and glucosylsphingosine (lysoGb-1) levels. Patients with type 3 GD had larger thyroid volumes and greater stiffness on SWE than patients with type 1 GD. GD patients also exhibited increased metabolic risk, including central obesity and elevated glucose levels. GD patients, despite normal thyroid hormone levels, exhibit subtle alterations in thyroid function indicators. Their increased risk of central obesity and glucose metabolism disorders, alongside higher TSH and FT4 levels, underscores the need for closer monitoring and further investigation.

## 1. Introduction

Thyroid diseases represent a significant health concern, affecting a substantial portion of the global population. The thyroid gland plays a crucial role in regulating metabolism, development, and growth. Any disruption in its function, including hypothyroidism or hyperthyroidism, can lead to a range of metabolic changes and complications. Lysosomal storage diseases can also secondarily result in thyroid dysfunction [1]. Gaucher disease (GD) is a rare genetic disorder caused by a deficiency in the enzyme glucocerebrosidase, leading to the accumulation of glucocerebroside in organs such as the spleen, liver, and bone marrow, which causes various systemic effects [2]. There is growing recognition that GD involves a complex interplay of downstream factors beyond simple substrate accumulation [3]. These include endoplasmic reticulum (ER) stress and the unfolded protein response (UPR), which disrupt protein folding and cellular homeostasis [3]. Calcium deregulation further contributes to cellular dysfunction, while mitochondrial impairment leads to energy deficits and oxidative stress [3]. Additionally, altered secretion and function of extracellular vesicles and heightened immunologic activity exacerbate inflammation, adding further complexity to the pathophysiology of GD [3]. The current literature on thyroid disorders in GD is limited, and there is a lack of comprehensive knowledge regarding the structure and function of the thyroid gland in both treated and untreated patients with GD [4,5,6]. Although direct thyroid involvement is not a typical feature of GD, patients may experience alterations in thyroid function due to the complex interplay between metabolic deregulation, mitochondrial dysfunction, systemic inflammation, and the potential influence of GD on the endocrine system.

## 2. Results

### 2.1. Thyroid Hormone Status of GD Patients

The median TSH level was 2.14 (µU/mL) (1.41–3.03) (Table 1). In four patients, TSH levels were observed to exceed 4.2 µU/mL (normal range: 0.27–4.2), leading to a diagnosis of hypothyroidism. No laboratory indicators of hyperthyroidism were observed in any of the patients. Isolated elevation of anti-TPO was noted in three subjects (all during imiglucerase treatment), while elevation of anti-Tg was observed in one additional subject. Elevation of both anti-TPO and anti-Tg was seen in one other subject.

The terms used are defined as follows: anti-TPO-anti-thyroid peroxidase antibodies; anti-Tg-anti-thyroglobulin antibodies; BMR-basal metabolic rate; BMI-body mass index; DBP-diastolic blood pressure; ERT- enzyme replacement therapy; estimated RMR-estimated resting metabolic rate; estimated TEE-estimated total energy expenditure; fat-fat tissue; fat-free mass-fat-free mass; FATM-fat mass; FT3-free triiodothyronine; FT4-free thyroxine; HDL-high-density lipoprotein cholesterol; INR-international normalized ratio; LDL-low-density lipoprotein cholesterol; neck BMD-neck bone mineral density; SBP-systolic blood pressure; SD-standard deviation; SRT- substrate reduction therapy; TG-triglycerides; TSH-thyroid stimulating hormone; total BMD-total bone mineral density; VAT mass-visceral adipose tissue mass; WC-waist circumference; WBC-white blood cell count.

### 2.2. Ultrasound and Elastography Examination of GD Subjects

Nodular goiter was observed in 14 patients (approximately 23% of the total). Three patients exhibited a focal lesion in the right lobe, three in the left lobe, and eight had nodular changes in both thyroid lobes. Significantly reduced echogenicity of the thyroid, along with the presence of hypoechoic areas and hyperechoic bands suggestive of chronic autoimmune inflammation, was noted in eight patients (n = 8/60; 13%) (Figure 1 and Figure 2). In contrast, isolated hyperechoic strands and a mild reduction in echogenicity were observed in 50 out of 60 patients, corresponding to approximately 83%. Several patients shows normal homogeneous echogenicity of the thyroid (Figure 3 and Figure 4).

### 2.3. Elastography Results

The results of thyroid gland elasticity measurements in patients with GD are presented in Table 1. Notably, the elasticity of the thyroid gland in GD falls within the normal range and does not deviate from the published norms for the general population.

### 2.4. Comparison Between GD Patients and Controls

The levels of TSH (2.14 µU/mL [1.41–3.03] vs. 1.64 µU/mL [1.26–2.35], *p* = 0.04) and FT4 (16.70 pmol/L [15.67–18.00] vs. 15.88 pmol/L [14.50–17.10], *p* = 0.02) were significantly higher in the GD group than in healthy individuals. There was no significant difference in thyroid volume between the patients and the control group. Patients with GD exhibited significantly higher WC (88 cm [82–95] vs. 77.5 cm [70–79], *p* < 0.001) and HC (96.3 cm [93–103] vs. 88.5 cm [80–96], *p* < 0.001) compared to the control group. Additionally, glucose and insulin levels were significantly higher in the GD group compared to the controls (glucose: 112 mg/dL [93–119] vs. 87 mg/dL [82–96], *p* < 0.001; insulin: 13.9 μU/mL [17–22.4] vs. 9.2 μU/mL [7.1–12.4], *p* = 0.02). Conversely, total cholesterol, LDL, and HDL levels were significantly lower in the GD group than in the controls (Table 2).

### 2.5. Comparison Between GD Patient Subgroups

The results of thyroid laboratory tests, as well as thyroid ultrasound and elastography, did not differ significantly between patients receiving ERT and those who were untreated.

In patients with nodular goiter (n = 14), significantly lower platelet (PLT) levels were noted (177 × 10^3^/µL [168–211] vs. 181 × 10^3^/µL [134–222], *p* < 0.001), along with higher chitotriosidase concentrations (341.3 nmol/mL/h [216.5–793.1] vs. 165.3 nmol/mL/h [94.8–670.6], *p* < 0.001) and higher lysoGb-1 levels (19.3 ng/mL [12.1–40.6] vs. 18.1 ng/mL [11.2–72.2], *p* < 0.001) compared to patients without nodular goiter. Importantly, no significant differences in age or BMI were observed between patients with and without nodular goiter.

Patients homozygous for c.1448T > C (p.L444P) with type 3 GD had a larger thyroid volume (15.8 [11.5–22.4] vs. 9.9 [7.3–13], *p* = 0.03), lower levels of anti-TPO antibodies (9 [7,8,9,10,11] vs. 12 [9,10,11,12,13,14,15,16,17,18,19], *p* < 0.05), and higher stiffness values (Q-Box mean) (25 [21–30] vs. 16.1 [11.8–20.9], *p* = 0.02). There was also a trend towards lower TSH levels in GD3 compared to GD1 patients (1.9 [1.2–2.6] vs. 2.3 [1.8–3.2], *p* = 0.06).

### 2.6. Correlation Analysis in GD Subjects

A positive correlation was observed between TSH levels and total BMD (r = 0.34, *p* = 0.02), as well as neck BMD (r = 0.40, *p* = 0.004). No correlation was found between TSH and body composition parameters, including estimated resting metabolic rate (RMR) and total energy expenditure (TEE).

Additionally, TSH and FT3 levels positively correlated with hemoglobin (Hg) concentration (r = 0.27, *p* = 0.04; r = 0.49, *p* < 0.05). A significant positive correlation was also observed between TSH levels and the PT index (r = 0.52, *p* < 0.01). Furthermore, the level of anti-Tg antibodies correlated with HC (r = 0.31, *p* = 0.02).

The Qbox mean for both the right and left thyroid lobes correlated with D-dimer levels (r = 0.40, *p* = 0.30 for the right lobe and r = 0.38, *p* = 0.04 for the left lobe) and Hg levels (r = −0.39, *p* = 0.03 for the right lobe and r = −0.44, *p* = 0.11 for the left lobe). No correlations were observed between TSH, FT3, FT4, and markers of GD, such as chitotriosidase, lysoGb-1, or ferritin levels.

## 3. Discussion

Metabolic and systemic dysfunctions caused by GD, including inflammation and hypermetabolism, could potentially affect thyroid function indirectly. Currently, there is limited data on the exact prevalence of thyroid diseases, as well as thyroid function and morphology in GD, and these data may vary depending on the population and region from which the patients originate. Regional differences in iodine status can lead to varying rates of thyroid disease, which should be considered when comparing data from different populations. Poland is classified as an area with adequate iodine supply following the implementation of universal salt iodization in 1997, which significantly improved iodine levels across the population. Recent studies confirm that Poland has maintained an adequate iodine intake, helping to prevent iodine-deficiency disorders such as goiter. In the Turkish study on 13 GD patients, 12 non-neuronopathic (GD1) and one subacute neuronopathic (GD3), no patients exhibited thyroid function abnormalities [4]. In a Langeveld et al. (Dutch data) study at baseline, 17 out of 22 patients (77%) exhibited normal thyroid hormone levels [5]. In the current study, most patients (93%) exhibited normal thyroid hormone levels. The risk of overt clinical hypothyroidism (7%) appears to be slightly higher than that of the general Polish population (approximately 1–6%) [7].

Compared to the control group, patients with GD exhibit higher levels of TSH and FT4. These elevations fall within the reference range, which may lead to them being overlooked during the management of GD patients. Previous studies did not analyze thyroid hormone levels in comparison to the control group [4,5]. The increased levels of free thyroid hormones could indicate hypermetabolism. However, as noted in Langeveld’s study, no correlation was found between metabolic measurements, including BMR, RMR, and TEE, and thyroid hormone levels in the current study [5]. It is plausible that changes in thyroid hormone levels are compensatory responses to other metabolic alterations. Physical and metabolic overload can indeed increase TSH levels. Conditions like obesity, dyslipidemia, hypertension, inflammation, metabolic syndrome, and prolonged physical stress are associated with elevated TSH, even when thyroid hormone levels are within normal ranges [8,9]. This connection indicates that TSH could serve as a novel marker for cardiometabolic risks [8]. Further studies are needed to assess thyroid hormone levels in GD in larger cohorts of patients, in comparison to healthy controls, and to establish the underlying causes and impact on the primary disease.

In the current study, there was no significant difference in thyroid laboratory tests between treated and untreated patients with GD. Long-term ERT resulted in a reduction in serum fT4 levels, and a decrease in REE was noted after several months of ERT treatment in the Langeveld et al. study [5]. In the current study, patients were assessed at a single time point; however, consistent with the observations of Langeveld, further research on the decline of fT4 levels is important to clarify the long-term effects of treatment and the dynamics of thyroid hormone levels in individuals with GD.

In the current study, patients with GD faced heightened metabolic risks, characterized by significantly larger WC and hip circumference, despite having a similar BMI to the control group. Additionally, glucose and insulin levels observed in GD patients were higher than those in controls. Lower HDL levels were also observed in GD subjects, consistent with previous reports linking low HDL to more severe advancement of GD [4,10]. This indicates that patients with GD remain at a higher metabolic risk, suggesting the need to monitor their WC, glucose metabolism, and lipid profile parameters. The literature on carbohydrate metabolism in GD is contradictory, but a few studies have shown an increased risk of insulin resistance and diabetes [11,12,13]. An increase in BMI has been reported in patients with GD during ERT [11]. After ERT, the prevalence of overweight was approximately 56–60%, attributed to decreased resting energy expenditure (REE) and inadequate calorie intake adjustment [10,11]. There is a lack of data regarding the risk of visceral obesity and the monitoring of WC in patients with GD.

A difference in thyroid volume, anti-TPO levels, and thyroid elasticity was observed between GD3 and GD1 patients. However, these changes remained within the reference range. The small sample size may have affected the results. To date, observations regarding thyroid hormone concentrations have primarily been reported for type 1 disease, aligning with the current results [4,5]. Further investigation is required to determine whether the increased thyroid stiffness and volume in GD3 patients hold clinical significance or are merely incidental findings.

The frequency of nodular goiter in GD, reported in the current study as 23%, is comparable to the prevalence of approximately 17–25% noted in a Polish publication. In the majority of patients (approximately 83%), isolated hyperechoic strands and a slight reduction in echogenicity were observed. These changes did not correlate with thyroid autoantibody levels (anti-TPO, anti-TG) and may be associated with a general inflammatory response. Further studies and close monitoring of thyroid ultrasound during treatment are essential in managing GD.

In the current study, 8% of patients had elevated levels of anti-TPO and/or anti-Tg antibodies. This finding aligns with data from the general Polish population, where the prevalence of TPOAb has been reported to range from 6.8% to 13% [14]. Interestingly, anti-TPO antibody levels were higher in the control group compared to GD patients. However, these changes were within the reference range and are likely of no clinical significance. In a study by Kilavuz et al. on Turkish GD patients, antithyroid antibodies were found in 4% (4/102) of cases: elevated anti-Tg in 2 GD1 patients, anti-TPO in 1 GD1 patient, and increased TRAb in 1 GD patient [15]. The differences in the prevalence of these antibodies may be due to geographical differences and varying iodine intake across regions.

In GD patients, the mean Q-box value for the right lobe was 17.2 kPa (range: 11.9–20.9 kPa), and for the left lobe, it was 14.9 kPa (range: 11.3–21.5 kPa). The mean elasticity of both lobes was 16.05 kPa. Notably, the elasticity of the thyroid gland in GD falls within the normal range and does not deviate from the published norms for the general population. In a study by Bojunga et al. (2012), it was determined that healthy thyroid tissue exhibits stiffness values in the range of approximately 10–20 kPa [16]. Additionally, a previous Polish publication reported that the mean elasticity of the thyroid gland in healthy controls was 16.18 ± 5.4 kPa [17].

However, glucocerebroside accumulation in the thyroid has not yet been confirmed. The normal elasticity of the thyroid gland in GD indirectly supports the theory of the absence of glucocerebroside accumulation in the thyroid. Metabolites in other metabolic storage disorders have been known to accumulate in the thyroid. Conditions such as amyloidosis, mucopolysaccharidoses, hemochromatosis, and glycogen storage disease may lead to various forms of thyroid dysfunction due to the accumulation of abnormal substances in thyroid tissue [18,19,20]. Therefore, while GD can have broad systemic effects, thyroid involvement and changes in thyroid echogenicity and structure observed on ultrasound appear to be more of a secondary or indirect effect rather than a result of direct lipid storage in the thyroid tissue itself.

Higher concentrations of GD progression markers, such as lysoGb-1 and chitotriosidase, along with lower platelet counts in patients with nodular goiter compared to those without nodular goiter, may hypothetically suggest an increased risk of thyroid nodule formation in patients with more advanced or less responsive to primary disease treatment [21]. Further studies on the risk of nodular goiter in patients with elevated chitotriosidase and/or lysoGb-1 levels are needed to better understand this potential association.

TSH has been shown to play a role in bone remodeling processes [22]. In the current observations, a positive correlation was observed between TSH levels and both total BMD and neck BMD (r = 0.34, *p* = 0.02), r = 0.40, *p* = 0.004, respectively). A positive correlation between TSH and BMD was also observed in studies on other populations, e.g., euthyroid postmenopausal women or women with type 2 diabetes [23,24].

Some researchers have indicated a potential relationship between TSH levels and Hg levels. Positive correlations were observed between TSH and FT3 levels and Hg concentration in GD patients in this study. While results vary in specific populations and conditions, some research shows, contrary to the current observations, that higher TSH levels, particularly in hypothyroid patients, may be linked to lower Hg levels, potentially due to the reduced metabolic activity in hypothyroidism [25,26,27]. This can result in mild anemia, which is more common in patients with overt hypothyroidism [25]. However, many studies report no significant direct correlation between TSH and Hg levels in the general population. The exact nature and clinical relevance of the TSH- relationship requires more focused studies in the GD population.

A positive correlation was also observed between TSH levels and the PT index, while thyroid stiffness on SWE was positively correlated with D-dimer levels. It is important to note that normal thyroid function is essential for proper coagulation, and some of the observed changes may represent adaptive responses.

Understanding the link between GD and thyroid function necessitates future studies involving larger cohorts. Careful monitoring of thyroid hormone levels in affected individuals is essential, as comprehensive management strategies to address thyroid disorders in GD are needed.

## 4. Materials and Methods

### 4.1. General Characteristics of GD Patients

The study included patients with GD recruited from across Poland and was conducted at the University Clinical Hospital in Poznań, Poland, between 2017 and 2023. The inclusion criteria were genetically confirmed GD1 or GD3 and informed consent to participate in the study. The exclusion criterion was a history of thyroidectomy, neck radiotherapy, and treatment with kinase inhibitors or corticosteroid therapy. Sixty patients with GD, with a median age of 37 years (IQR 27–47), were recruited for the study, including 48 with type 1 and 12 with type 3. The study group consisted of 30 women and 30 men. At the time of the study, 46 patients were receiving enzyme replacement therapy (ERT), comprising four treated with velaglucerase and 42 with imiglucerase. Two patients were undergoing substrate reduction therapy (SRT), while 12 patients were not receiving any treatment. The control group consisted of 46 healthy individuals without chronic diseases who were not taking any regular medications and were matched for age, sex, and BMI with the study group.

GD patients were classified into genotype groups as follows: nine patients with homozygous c.1226A > G (p.N370S), 12 patients with homozygous c.1448T > C (p.L444P), ten patients with heterozygous c.1226A > G/c.1448T > C (p.N370S/p.L444P), 23 patients with heterozygous c.1226A > G/another variant (p.N370S/other variant), two patients with heterozygous c.1448T > C/another variant (p.L444P/other variant), and four patients with other genotypes.

### 4.2. Thyroid Ultrasound and Shear Wave Elastography (SWE)

Conventional ultrasound (US) and Shear Wave Elastography (SWE) were performed using the AIXPLORER system by Supersonic Imagine, equipped with a 2–10 MHz linear transducer. Examinations were conducted by two experienced sonographers (MR and ESP). Thyroid volume was calculated using the formula: thyroid volume (mL) = (π/6) × length × width × depth (cm). For each thyroid lobe, the length, width, and depth were measured in centimeters (cm), and the total thyroid volume was obtained by summing the volumes of both lobes. The isthmus was not included in the volume calculation. Thyroid gland elasticity was assessed both qualitatively and quantitatively.

By measuring the propagation velocity of the shear waves at every point of the image, an elasticity map could be deduced. This included color-coded displays depicting tissue stiffness using a color scale from blue (soft) through green and yellow (medium elasticity) to red (hard). The classification of the affected thyroid parenchyma in a 4-point modified Ueno scale was used to describe tissue stiffness, where elasticity score (ES) I meant completely normal thyroid elasticity (blue color); ES II meant parenchyma of predominantly normal elasticity; however, containing areas of slightly increased stiffness (green and yellow color); ES III meant parenchyma of intermediate elasticity, where yellow color predominates and may contain some tiny areas of highly increased stiffness, depicted in red; and ES IV meant most of the thyroid parenchyma presents highly increased stiffness, with predominating red color (5). The quantitative information was depicted as a stiffness index and expressed in kilopascals on a continuous scale. Tissue stiffness was quantified by Young’s Module E, which corresponded to the speed of propagation of the shear wave. For the quantitative assessment, stiffness values were measured in kilopascals (kPa) and recorded as both maximum (Q-box max) and mean (Q-box mean) values.

### 4.3. Anthropometric, Biochemical and Hormonal Assessment

Anthropometric measurements, including body weight, height, waist circumference (WC), and hip circumference (HC), were taken. Body mass index (BMI) was calculated using the standard formula: BMI = weight (kg)/height (m²).

Venous blood samples were obtained from each participant following standard phlebotomy protocols. A complete blood count, including white blood count (WBC), monocyte, and lymphocyte counts, was assessed using a Sysmex XN-1000 hematology analyzer. Glucose measurements were performed in serum by the hexokinase method with a coefficient of variation (CV) of ≤3%.

High-sensitivity C-reactive protein (hsCRP), insulin, ferritin, TSH, FT3, FT4, anti-thyroid peroxidase (anti-TPO), and anti-thyroglobulin (anti-Tg) were measured using a Cobas 6000, with assays provided by Roche Diagnostics. Total cholesterol, high-density lipoprotein cholesterol (HDL), and triglycerides were evaluated by the enzymatic colorimetric method. Low-density lipoprotein cholesterol (LDL) was estimated using the Friedewald formula: LDL = total cholesterol−HDL−(Triglycerides/5). Coagulation parameters, including international normalized ratio (INR), prothrombin time (PT), and D-dimer levels, were analyzed according to established methodologies. The chitotriosidase activity was assessed in serum by a modified method described by Hollak et al. [28]. The concentration of glucosylsphingosine (LysoGb-1) was determined using an adapted liquid chromatography–tandem mass spectrometry method described by Staufer et al. [29]. The measurements of both parameters were supported by the Archimedlife company.

### 4.4. Body Composition Analysis and Densitometry

Patient body composition was assessed using air plethysmography via BodPod (Life Measurement Inc., Concord, CA, USA), and measurements were performed according to the validated protocol. Participants came to sessions after overnight fasting. The patients were advised to wear approved clothing, such as bathing suits, compression shorts, and bras with no wiring or padding, and not to wear any jewelry. Additionally, each patient wore a swim cap to decrease hair volume. The equipment was calibrated every morning before study sessions; each session took place at 21–26 °C with relative humidity between 20 and 70% [30,31,32].

All study participants had a dual-energy X-ray absorptiometry (DXA) scan at the Department of Human Nutrition and Dietetics by the same technician on a GE Lunar Prodigy machine (General Electrics Healthcare Medical Systems, Europe, Belgium). The DXA measured bone mineral content (BMC, g), bone mineral density (BMD, g/cm^2^), and body composition, including visceral adipose tissue assessment (VAT). Scans were performed in the morning in a fasted state. The quality control was performed according to the user manual on each day of the study visits. Subjects had to remove all metal parts of clothing and accessories [31].

### 4.5. Statistical Analysis

Descriptive statistics summarized demographic, clinical, and laboratory data. For normally distributed variables, mean and standard deviation were presented, while non-normal variables were reported using median and interquartile range. Pearson’s correlation was applied for interval scale variables, and the Chi-square test for categorical variables. Cohen’s kappa coefficient assessed interobserver agreement. Statistical significance was set at *p* < 0.05. Student’s t-test or the Mann–Whitney test was used for comparisons, depending on distribution normality. Statistical analysis was performed using Statistica version 14.

## 5. Conclusions

Patients with GD exhibit a metabolic risk profile characterized by higher TSH and FT3 levels within normal ranges, along with elevated glucose and insulin levels and lower HDL levels. Additionally, patients with GD appear to have a risk of nodular goiter and chronic autoimmune thyroiditis that is comparable to that of the general population. Patients with GD exhibit normal thyroid gland elasticity values. The observations regarding elevated levels of chitotriosidase and lysoGb-1 in patients with nodular goiter are particularly intriguing, and further studies are needed to explore this association. Advanced research is necessary to elucidate the relationship between GD and thyroid disorders in these patients.

## Figures and Tables

**Figure 1 ijms-25-13636-f001:**
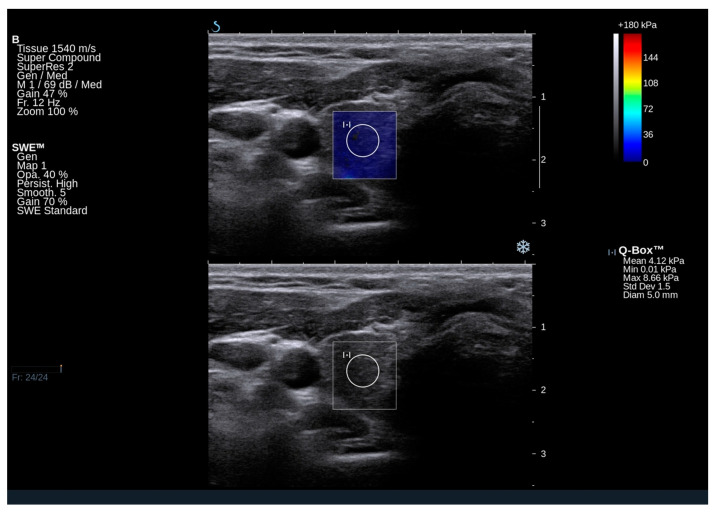
An ultrasound image of the right thyroid lobe in a 30-year-old patient with Gaucher disease type 3, demonstrating normal elasticity of the thyroid parenchyma. Elasticity was assessed on both qualitative and quantitative scales. The conventional grayscale image shows heterogeneous decreased echogenicity of the thyroid and a few white lines indicative of fibrosis.

**Figure 2 ijms-25-13636-f002:**
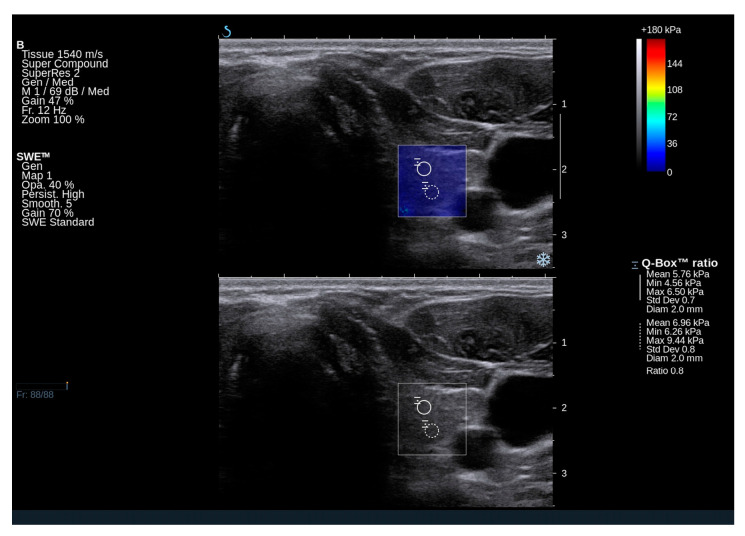
An ultrasound image of the left thyroid lobe in a 30-year-old patient with Gaucher disease type 3, demonstrating normal elasticity of the thyroid parenchyma. Elasticity was assessed on both qualitative and quantitative scales. The conventional grayscale image shows heterogeneous decreased echogenicity of the thyroid and a few white lines indicative of fibrosis.

**Figure 3 ijms-25-13636-f003:**
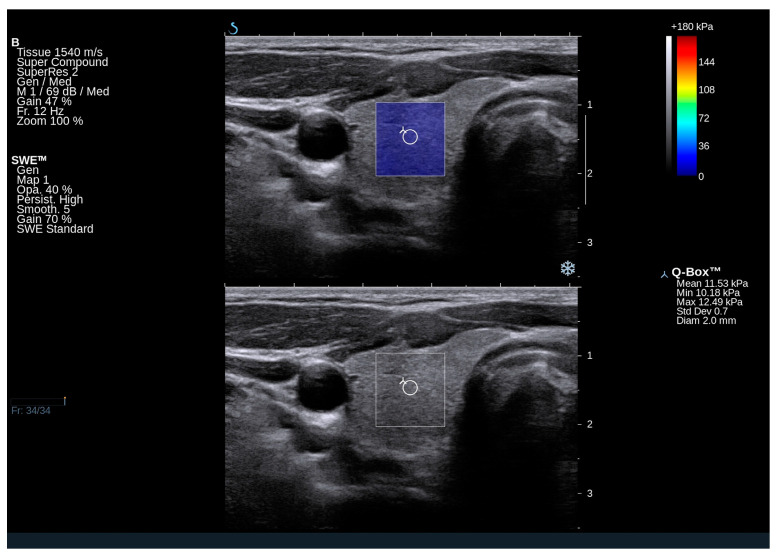
An ultrasound image of the right thyroid lobe in a 32-year-old patient with Gaucher disease type 1, demonstrating normal elasticity of the thyroid parenchyma. Elasticity was assessed on both qualitative and quantitative scales. The conventional grayscale image shows normal homogeneous echogenicity of the thyroid.

**Figure 4 ijms-25-13636-f004:**
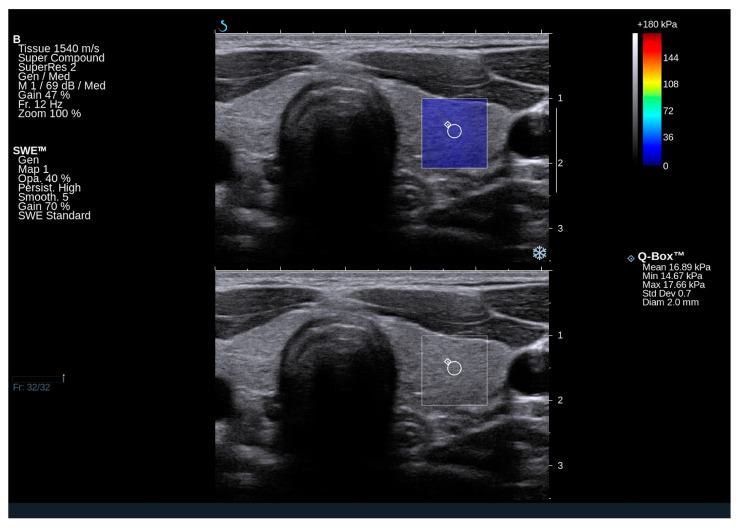
An ultrasound image of the left thyroid lobe in a 32-year-old patient with Gaucher disease type 1, demonstrating normal elasticity of the thyroid parenchyma. Elasticity was assessed on both qualitative and quantitative scales. The conventional grayscale image shows normal homogeneous echogenicity of the thyroid.

**Table 1 ijms-25-13636-t001:** Characteristics of the studied Gaucher disease (GD) group, including clinical, hormonal, ultrasound, anthropometric, biochemical, body composition, and bone density parameters.

Variable	Results (Median (Interquartile Range) or Number)	Reference Range
Sex	30 males; 30 females	
Type of GD	48 type 1; 12 type 3	
Type of treatment	ERT n = 46; SRT n = 2 Treatment-naïve n = 12	
Cumulative dose of treatment * (U)	240 (75–458)	
Time of treatment (years)	13 (4–19)	
Genotypes	Homozygous c.1226A > G (p.N370S) n = 9 Homozygous c.1448T > C (p.L444P) n =12 c.1226A > G/c.1448T > C (p.N370S/p.L444P) n = 10, c.1226A > G/another variant (p.N370S/other variant) n = 23 p.L444P/other variant n = 2 Other genotypes n = 4	
**GD monitoring parameters**		
Chitotriosidase (nmol/mL/h)	212.9 (99.3–743.3)	<150
LysoGb-1 (ng/mL)	18.3 (11.4–50)	0–14
**Anthropometrics**		
Age (years)	37 (27–47)	
BMI (kg/m^2^)	25.2 (22.2–27.1)	
Waist circumference (cm)	88 (82–95)	
Hip circumference (cm)	96.3 (93–103)	
Systolic blood pressure (SBP) (mmHG)	127.5 (113–149)	
Diastolic blood pressure (SBP)	81.5 (76–87)	
**Thyroid function and morphology**		
TSH (µU/mL)	2.14 (1.41–3.03)	0.27–4.2
FT3 (pmol/L)	5.21 (4.65–5.88)	3.9–6.7
FT4 (pmol/L)	16.70 (15.67–18.00)	11.5–21
Anti-TPO (IU/mL)	10 (<34)	<34
Anti-Tg (IU/mL)	12 (10–115)	10–115
Thyroid volume (mL)	10.8 (8.0–15.9)	
Right thyroid lobe volume (mL)	6.2 (4.8–9.4)	
Left thyroid lobe volume (mL)	4.7 (3.5–7.0)	
Right lobe Qbox mean (kPa)	17.2 (11.9–20.9)	
Right lobe Qbox min (kPa)	11.3 (6.8–13.3)	
Right lobe Qbox max (kPa)	25.4 (18.8–31.5)	
Right SD	3.5 (2.7–4.5)	
Left lobe Qbox mean (kPa)	14.9 (11.3–21.5)	
Left lobe Qbox min (kPa)	8.7 (5.3–12.5)	
Left lobe Qbox max (kPa)	22.9 (17.3–29.6)	
Left SD	3.2 (2.1–4.0)	
**Biochemical parameters**		
Glucose (mg/dL)	112 (93–119)	70–99
Insulin (µU/mL)	13.9 (17–22.4)	2–25
Total cholesterol (TC)	144 (127–177)	115–190
Low-density cholesterol (LDL)	84 (70–97.6)	<115
High-density cholesterol (HDL)	47 (34–62)	>35 in men; >45 in women
Triglycerides (TG)	75 (62–163)	65–150
Ferritin (ng/mL)	156 (69–406.5)	22–275
**Coagulation parameters and blood count**		
INR	1.11 (1.05–1.15)	0.86–1.14
PT (s)	12.1 (11.5–12.6)	9.4–12.5
PI (%)	90 (87–95)	87–116
D-dimer	0.28 (0.18–0.45)	<0.5
White blood count (WBC) (10^3^/µL)	5.1 (4.3–6.0)	3.6–11.0
Hemoglobin (g/dL)	13.8 (12.7–15.5)	11.8–15.8
PLT (10^3/^µL)	178 (124.5–212)	130–400
**Body composition**		
Fat (%)	32.4 (26.3–40.0)	
Fat-free mass (%)	67.6 (60–73.7)	
Estimated RMR (kcal/day)	1258 (1106–1529)	
Estimated TEE (kcal/day)	1928.5 (1681.0–2327.0)	
BMR (kcal/day)	5977 (6644–7255)	
**Densitometry**		
Total BMD (g/cm^2^)	1.105 (1.047–1.183)	
Neck BMD (g/cm^2^)	0.984 (0.874–1.067)	
L1-L4 BMD (g/cm^2^)	1.078 (1.013–1.188)	
VAT mass (g)	501.4 (352.1–1061.2)	

* Calculated per body weight (cumulative dose calculated as the product of the dose per kg of body weight and the duration of therapy in years).

**Table 2 ijms-25-13636-t002:** Comparison of anthropometric, hormonal, ultrasound, and biochemical parameters between GD patients and healthy subjects.

Variable	GD Patients (Median (Interquartile Range) or Number (Percentage))	Controls (Median (Interquartile Range) or Number (Percentage))	*p* Value
Age (years)	37 (27–47)	37 (26.4–47.5)	NS
BMI (kg/m^2^)	25.2 (22.2–27.1)	23.9 (21.8–26.7)	NS
Waist circumference (cm)	88 (82–95)	77.5 (70–79)	<0.001
Hip circumference (cm)	96.3 (93–103)	88.5 (80–96)	<0.001
TSH (µU/mL)	2.14 (1.41–3.03)	1.64 (1.26–2.35)	0.04
FT3 (pmol/L)	5.21 (4.65–5.88)	5.06 (4.63–5.61)	NS
FT4 (pmol/L)	16.70 (15.67–18.00)	15.88 (14.50–17.10)	0.02
AntiTPO (IU/mL)	10 (<34)	13 (11–16)	0.02
AntiTg (IU/mL)	12 (10–115)	10 (10–15)	NS
Thyroid volume (mL)	10.8 (8.0–15.9)	11.5 (8.4–15.2)	NS
Right thyroid lobe volume (mL)	6.2 (4.8–9.4)	6.11 (4.7–8.5)	NS
Left thyroid lobe volume (mL)	4.7 (3.5–7.0)	5.2 (3.6–6.5)	NS
Glucose (mg/dL)	112 (93–119)	87 (82–96)	<0.001
Insulin (µU/mL)	13.9 (17–22.4)	9.2 (7.1–12.4)	0.02
Total cholesterol (TC)	144 (127–177)	182	0.001
Low-density cholesterol (LDL)	84 (70–97.6)	102.4	0.004
High-density cholesterol (HDL)	47 (34–62)	55	0.03
Triglycerides (TG)	75 (62–163)	94	NS

Anti-TPO-anti-thyroid peroxidase antibodies; anti-Tg-anti-thyroglobulin antibodies; BMI-body mass index; FT3-free triiodothyronine; FT4-free thyroxine; HDL-high-density lipoprotein cholesterol; LDL-low-density lipoprotein cholesterol; TG-triglycerides; TSH-thyroid stimulating hormone.

## Data Availability

The datasets used and/or analyzed during the current study are available from the corresponding author upon reasonable request.

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
