# Peer review of "Thyroid Function and Morphology in Gaucher Disease: Exploring the Endocrine Implications"

_ijms, 2024, doi:10.3390/ijms252413636_

Round 1
Reviewer 1 Report
Comments and Suggestions for Authors
Dear Authors,
Thank you very much for submitting such an interesting paper to the IJMS. Below I attach some comments and suggestions regarding this paper:
- The abstract seems to be too long – too many words are included in this form and I suppose that the section of results is too evaluated as for the abstract. Please shorten the abstract and adjust it according to the guidelines for authors
- I would rather recommend changing the order of the manuscript a little bit and adding the materials and methods section just after the introduction section
- In the materials and methods section please indicate from where, which hospital or clinic were the patients recruited? Besides, I would recommend adding a table summarizing the characteristics of the patients involved in the study, it will be easier for the readers to read it with a table. Also, how many male and female patients were included in the study? What was the mean and median age of the patients? What were the inclusion and exclusion criteria in this study regarding the patient recruitment? Please specify it as well.
- Line 336 and 213 – there is a double space. Please remove it
- If it is possible please check the paper once again in terms of English since I have found some minor grammatical errors
Congratulations on your very interesting and important work and good luck with your further research
Author Response
Comments 1: The abstract seems to be too long – too many words are included in this form and I suppose that the section of results is too evaluated as for the abstract. Please shorten the abstract and adjust it according to the guidelines for authors
Response1: The abstract has been condensed; thank you for the suggestion. It now appears clearer and more concise.
Comments 2: would rather recommend changing the order of the manuscript a little bit and adding the materials and methods section just after the introduction section
Response 2: The order of the methodology section was based on the journal's suggestions and template structure. However, if the editor has no objections, we can certainly place the Materials and Methods section immediately after the introduction, before the results.
Comments 3: The materials and methods section please indicate from where, which hospital or clinic were the patients recruited? Besides, I would recommend adding a table summarizing the characteristics of the patients involved in the study, it will be easier for the readers to read it with a table. Also, how many male and female patients were included in the study? What was the mean and median age of the patients? What were the inclusion and exclusion criteria in this study regarding the patient recruitment? Please specify it as well.
Response 3: The study included patients with Gaucher disease recruited from across Poland and was conducted at the University Clinical Hospital in Poznań, Poland, between 2017 and 2023. The study group consisted of 30 women and 30 men. The inclusion criteria were genetically confirmed Gaucher disease type 1 or 3 and informed consent to participate in the study.The exclusion criterion was a history of thyroidectomy, neck radiotherapy, treatment with kinase inhibitors, or corticosteroid therapy. The information including median age and above mentioned facts has been included in the methodology section. The suggested data have been added to Table 1 to ensure that all essential, relevant information on patienrs characteristics is clearly accessible in one place. Thank you for the suggestion.
Comments 4: Line 336 and 213 – there is a double space. Please remove it
Response 4: The line spacing has been adjusted
Comments 5: If tt is possible please check the paper once again in terms of English since I have found some minor grammatical errors
The text was corrected by English native speaker - details in acknowledgment section.

Reviewer 2 Report
Comments and Suggestions for Authors
I had the pleasure of reviewing the work of Kaluzna et al. This study found that thyroid-stimulating hormone (TSH) and free triiodothyronine (FT3) levels were significantly higher in Gaucher disease (GD) patients compared to controls, although both remained within the normal range. I have a few comments aimed at improving the manuscript.
Firstly, I believe the cohort of GD patients could be better characterized, perhaps with the addition of a new table that includes details such as age, the duration of treatment, measurements of lyso-Gb1, and chitotriosidase levels. This additional information will be useful for the later evaluation of thyroid function in relation to these variables.
Minor comments:
- Table 1: Please include the reference range for the data.
- Table 2: There are two references listed as [7].
- Line 123: Lyso-Gb1 levels were considered elevated; however, they are within the expected range for GD patients under treatment. It would be helpful to clarify this, noting the importance of treatment duration and patient age in interpreting these levels. Please refer to PMID: 32998334 for further context.
I hope my suggestions will help improve the manuscript.
Author Response
Comments 1: Firstly, I believe the cohort of GD patients could be better characterized, perhaps with the addition of a new table that includes details such as age, the duration of treatment, measurements of lyso-Gb1, and chitotriosidase levels. This additional information will be useful for the later evaluation of thyroid function in relation to these variables.
Response 1: The suggested data are included in Table 1, with clinical data from the medical history added at the top. Information on disease markers, such as lysoGL1 and chitotriosidase were included. Data on duration of treatment and cumulative dose of treatment ERT were also included in table.
Comments 2: Table 1: Please include the reference range for the data.
Response 2: The reference ranges were added to the Table S1 (Supplement).
Comment 3: Table 2: There are two references listed as [7].
Response 3: Thank you, the references were removed
Comment 4: Line 123: Lyso-Gb1 levels were considered elevated; however, they are within the expected range for GD patients under treatment. It would be helpful to clarify this, noting the importance of treatment duration and patient age in interpreting these levels. Please refer to PMID: 32998334 for further context. I hope my suggestions will help improve the manuscript.
Response 4: We are grateful for your comments which improved the manuscript, the text fragment about Lyso-Gb1 was corrected and the appropriate citation was added. The concentration of this biomarker was higher in patients with nodular goiter than in patients without goiter, but it was not elevated

Round 2
Reviewer 2 Report
Comments and Suggestions for Authors
Dear Authors,
I have the pleasure of reviewing your work again. However, I noticed that several review comments, such as those on lines 54 and 243, are still visible in the document. It seems that the wrong version might have been submitted.
Additionally, I suggest that the reference values be included in Table 1 itself, rather than as supplementary material. This would make the table more comprehensive and sufficient on its own.
Author Response
Comments 1: I have the pleasure of reviewing your work again. However, I noticed that several review comments, such as those on lines 54 and 243, are still visible in the document. It seems that the wrong version might have been submitted.
All changes in manuscript were accepted and all comments were removed.
Comments 2: Additionally, I suggest that the reference values be included in Table 1 itself, rather than as supplementary material. This would make the table more comprehensive and sufficient on its own.
The reference ranges were added to Table 1.
